# Economic and social impacts of COVID-19 and public health measures: results from an anonymous online survey in Thailand, Malaysia, the UK, Italy and Slovenia

Anne Osterrieder [1,2] Giulia Cuman [3] Wirichada Pan-Ngum [1,4] Phaik Kin Cheah,[5] Phee-Kheng Cheah,[6,7] Pimnara Peerawaranun,[1] Margherita Silan,[8] Miha Orazem,[9,10] Ksenija Perkovic [11] Urh Groselj [9,12] Mira Leonie Schneiders,[1,2,13] Tassawan Poomchaichote,[1] Naomi Waithira,[1,2] Supa-at Asarath,[1] Bhensri Naemiratch,[1] Supanat Ruangkajorn,[1] Lenart Skof [14] Natinee Kulpijit,[1] Constance R S Mackworth-Young [15] Darlene Ongkili,[16] Rita Chanviriyavuth,[1] Mavuto Mukaka,[1,2] Phaik Yeong Cheah [1,2,13]

For numbered affiliations see end of article.

**Correspondence to**
Phaik Yeong Cheah;
Phaikyeong@tropmedres.ac

## ABSTRACT

**Objectives** To understand the impact of COVID-19 and public health measures on different social groups, we conducted a mixed-methods study in five countries ('SEBCOV—social, ethical and behavioural aspects of COVID-19'). Here, we report the results of the online survey.

**Study design and statistical analysis** Overall, 5058 respondents from Thailand, Malaysia, the UK, Italy and Slovenia completed the self-administered survey between May and June 2020. Poststratification weighting was applied, and associations between categorical variables assessed. Frequency counts and percentages were used to summarise categorical data. Associations between categorical variables were assessed using Pearson's $\chi^2$ test. Data were analysed in Stata 15.0

**Results** Among the five countries, Thai respondents reported having been most, and Slovenian respondents least, affected economically. The following factors were associated with greater negative economic impacts: being 18–24 years or 65 years or older; lower education levels; larger households; having children under 18 in the household and and having flexible/no income. Regarding social impact, respondents expressed most concern about their social life, physical health, mental health and well-being.

There were large differences between countries in terms of voluntary behavioural change, and in compliance and agreement with COVID-19 restrictions. Overall, self-reported compliance was higher among respondents who self-reported a high understanding of COVID-19. UK respondents felt able to cope the longest and Thai respondents the shortest with only going out for essential needs or work. Many respondents reported seeing news perceived to be fake, the proportion varying between countries, with education level and self-reported levels of understanding of COVID-19.

## Strengths and limitations of this study

► Our research findings help to address an evidence gap as identified by the global research community in a recent study on COVID-19 research priorities, which identified public health messaging, compliance and trust in public health interventions and evaluation of these interventions in varied settings as areas of high priority (BMJ Global Health Vol 5, Issue 7 (https://gh.bmj.com/content/5/7/e003306).

► Because we recruited a reasonably large sample size in each country (between 700 and 1400), we were able to compare population segments (eg, men vs women, younger vs older people, those with lower vs higher levels of education) in the whole cohort, and between countries.

► Our online survey enabled us to capture people's experiences and concerns in multiple domains, in five countries, all of which had restrictions in place, during the relatively early stage of the COVID-19 pandemic.

► We did not aim to obtain nationally representative samples and acknowledge that although we used weighting strategies in our analysis, our results may not be fully representative of the populations in the respective countries.

► Our study captured the views and perceptions of respondents on the socioeconomic impact of COVID-19 public health measures, rather than data on standard indicators of economic and social impacts.

**Conclusions** Our data showed that COVID-19 and public health measures have uneven economic and social impacts on people from different countries and social groups. Understanding the factors associated with these

impacts can help to inform future public health interventions and mitigate their negative consequences.

**Trial registration number** TCTR20200401002.

## INTRODUCTION

COVID-19 is a respiratory disease caused by the novel coronavirus SARS-CoV2, which is transmitted through droplets, close contact and aerosols.[1 2] The SARS-CoV2 outbreak was first reported in December 2019 in Wuhan, China,[3] with the WHO declaring it Public Health Emergency of International Concern on 30 January 2020 and a global pandemic on 11 March 2020.[1]

In the absence of widely available vaccines and pharmaceutical treatments, the impact of COVID-19 is being mitigated using non-pharmaceutical interventions (NPIs).[4 5] Examples of NPIs include social distancing (or 'physical distancing') measures, such as isolation of sick individuals, quarantine of exposed individuals, contact tracing, voluntary shielding, travel-related restrictions and personal protective measures, such as hand hygiene and wearing face masks.[4 6 7] Scientific evidence indicates that NPIs are effective measures to contain a pandemic and ease pressures on healthcare systems.[6–12] However, authorities and policymakers need to consider the societal, economic and ethical impacts of these public health measures, in particular on vulnerable groups.[13 14] Such groups might be disproportionally affected by NPIs and/or might be unable to comply with them,[15] for example, due to loss of income when having to isolate at home, crowded living conditions,[14] or not being able to afford masks.[16]

As the COVID-19 pandemic continues, evidence is urgently needed to understand how people perceive and experience NPIs, which groups are disproportionally negatively affected by NPIs, and how communication is perceived by various social groups.[17] These data can be used to supplement standard indicators of economic and social impacts to provide a better understanding of the effects of COVID-19 and its related public health measures. This understanding is important so that the policies can be improved to minimise the negative impact of COVID-19 on people's lives, and to improve communications.

Here, we report the highlights of an online survey conducted in Southeast Asia (Thailand and Malaysia, both upper middle-income countries), and Europe (the UK, Italy and Slovenia, all high-income countries) between 1 May to 30 June 2020 as part of the mixed-methods study 'Social, ethical and behavioural aspects of COVID-19' (SEBCOV).[18] These findings help to address an evidence gap as identified by the global research community in a recent study on COVID-19 research priorities,[19] which identified public health messaging, compliance and trust in public health interventions, and evaluation of these interventions in varied settings as areas of high priority.[19]

## METHODS

### Study area

The survey was conducted in five countries (population in 2020 indicated in brackets:[20] Thailand (69.8 million) and Malaysia (32.4 million) in Southeast Asia; and UK (67.9 million), Italy (60.5 million) and Slovenia (2.1 million) in Europe.

### Survey development

The survey contained five sections with 36 questions (single-answer multiple choice and five-point Likert scales) on (1) sociodemographic information; (2) income, occupation status and economic impacts of COVID-19 restrictions; (3) sources of, preferences and perceptions regarding COVID-19-related communication, and the occurrence of 'fake news' (untrue information presented as news) and (4) perceived levels of understanding of COVID-19 and NPIs, agreement with NPIs, voluntary behavioural changes and concerns and coping strategies relating to restrictions.[21] The Malaysia and UK surveys were administered in English, with the other surveys translated into the respective country languages. The self-administered online survey was set up using the 'JISC Online surveys' platform.[22]

### Patient and public involvement

The survey questions were pilot-tested with 25 people from participating countries, and revised accordingly based on feedback. In addition, the Bangkok Health Research Ethics Interest Group, a public involvement group set up by the Mahidol Oxford Tropical Medicine Research Unit,[23] discussed the study and the survey questions in a dedicated virtual meeting. Selected questions were tested using an adapted cognitive testing technique using the 'thinking out loud' approach,[24] and the collaborative virtual sticky notes board 'Padlet'.[25]

### Participant selection and recruitment

Adults of any age residing in Thailand, Italy, Malaysia, UK or Slovenia at the time of the study were eligible to take part. Participants needed to be able to use a computer or smart phone to access the survey and provide online consent to participate.

The survey was open from 1 May to 30 June 2020 (1–30 June for Slovenia due to late start). Participants were recruited using a combination of approaches: snowball sampling through personal and professional networks (via email, social media and messenger apps, mailing lists and organisations such as the Medical Chamber[26] in Slovenia); a polling company[27] in Thailand and through promoted posts on Facebook. Facebook allows users to 'boost' posts to selected demographic audiences for a small fee, so that the post appears on their Facebook newsfeed.[28] To achieve more balanced responses in the categories of gender, education level and geographic distribution, promoted Facebook posts were targeted at people with primary or lower/secondary education in UK and Malaysia; potential participants in Wales, Scotland

and Northern Ireland in the UK and at men in the UK and Italy.

## Sample size

Each country aimed to recruit a minimum sample of 600 respondents, exceeding the 40–200 respondents recommended for a mixed-methods study.[29] A minimum sample size of 600 respondents is adequate to estimate the prevalence of a response assuming a 50% prevalence rate, with 95% confidence and with a precision of 4%. The 50% prevalence is the standard assumption for precision sample size calculations when the true prevalence is not available, as this gives the highest sample size for a binomial distribution for a desired level of precision. The following sample size formula was used n= $(Z^2_{1-\alpha/2}*P(1-P))/d^2$ where P is the anticipated prevalence, d is the margin of error, $Z_{1-a/2}$ is the standard normal value corresponding to the upper tail probability of α/2, α=0.05 (for a 95% CI) and n is the sample size.

## Statistical analysis

To simplify analysis, answers in the following categories were combined as follows: 'slightly agree/highly agree' were combined into one 'agree', category and 'slightly/strongly disagree' responses into one 'disagree' category. To understand the distribution of the basic demographic variables in the respondent sample, the observed frequencies and sample characteristics are reported using unweighted percentages (online supplemental table 1). The characteristics for the rest of the variables are presented using the observed survey frequency counts followed by weighted percentages (online supplemental tables 2–37). Poststratification weighting was used to align the composition of the respondents' sample with the known distribution of the whole population's characteristics, reducing sampling error. Weights were computed considering three stratifying variables that were available from population census data from each country,[30] namely, gender, age and education level. Weights were obtained as the ratio between the proportion of each possible combination of the three variables in the whole country population and the correspondent proportion in the respondent sample.

Survey data were analysed using Stata V.15.0 software.[31] Frequency counts and percentages were used to summarise categorical data. Associations between categorical variables were assessed using Pearson's $\chi^2$ test. P values have been provided in the tables and considered statistically significant below the two-sided alpha=0.05 level. All p values presented in the tables are for global tests of significance. Practical significance was taken into account when interpreting differences in the results.

## RESULTS

At the time of the inception of this study, governments in Thailand, Malaysia, Italy, the UK and Slovenia had initiated public health measures, using varying degrees

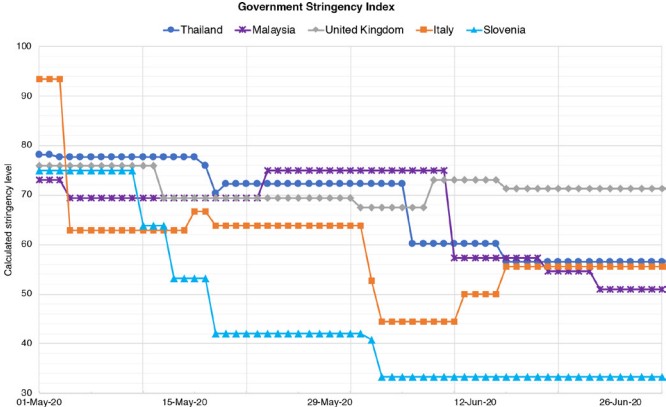

**Figure 1** Government stringency indices in Thailand, Malaysia, UK, Italy and Slovenia between 1 May and 30 June 2020. A higher score indicates a stricter government response, that is, 100 = strictest.[31]

of 'lockdowns' to curb the pandemic. Figure 1 shows a visualisation of the 'Stringency Index' (SI) of the public health responses of the five government over the study period, drawing on data provided by the Oxford COVID-19 Government Response Tracker (OxCGRT).[32] The OxCGRT tool tracks government policies and interventions from more than 180 countries on standardised indicators, and aggregates the data into a 'SI' for each country on a scale from 0 to 100, with 100 indicating the strictest response.[32] For example, Italy had the strictest public health measures in early May (SI=93) and then gradually lifted and reintroduced restrictions, whereas restrictions in the UK remained at around the same level (SI=69–76) throughout the study period. Restrictions in Slovenia were substantially eased from June onwards (SI=33).

## Characteristics of survey respondents

A total of 5058 participants took part in the survey: 1476 respondents from Thailand (29%), 827 from Malaysia (16%), 1009 from the UK (20%), 712 from Italy (14%) and 1034 from Slovenia (20%; online supplemental table 1, unweighted data). Overall, around 40% identified as male, around 60% as female and around 1% as 'other/prefer not to say'. Of all respondents, 8% were 18–24 years old, 17% were aged 25–34 years old, 65% were 35–64 years old and 10% fell into the 65+ age group. Overall, 33% had primary or lower (from here on referred to as 'primary') or secondary education, whereas 67% had tertiary education. Overall, 21% of respondents lived in large households with five or more people. A total of 59% of respondents received a fixed income (salary/benefits/pension), 31% had flexible income (contract and freelance) and 10% received no or 'other income'. Overall, 36% lived with children under 18 years in their household, and 29% reported that they or a household member belonged to a 'vulnerable group' (persons aged 70 or older, pregnant women or people with serious health conditions). Finally, 19% were healthcare provider/workers. Online supplemental table 1 provides

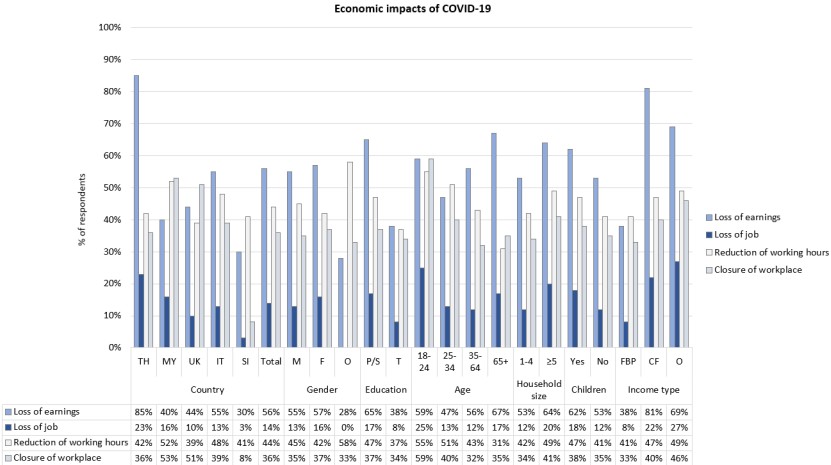

| | Country | | | | | | Gender | | | Education | | Age | | | | Household size | | Children | | Income type | | |
|---|---|---|---|---|---|---|---|---|---|---|---|---|---|---|---|---|---|---|---|---|---|---|
| | TH | MY | UK | IT | SI | Total | M | F | O | P/S | T | 18-24 | 25-34 | 35-64 | 65+ | 1-4 | ≥5 | Yes | No | FBP | CF | O |
| Loss of earnings | 85% | 40% | 44% | 55% | 30% | 56% | 55% | 57% | 28% | 65% | 38% | 59% | 47% | 56% | 67% | 53% | 64% | 62% | 53% | 38% | 81% | 69% |
| Loss of job | 23% | 16% | 10% | 13% | 3% | 14% | 13% | 16% | 0% | 17% | 8% | 25% | 13% | 12% | 17% | 12% | 20% | 18% | 12% | 8% | 22% | 27% |
| Reduction of working hours | 42% | 52% | 39% | 48% | 41% | 44% | 45% | 42% | 58% | 47% | 37% | 55% | 51% | 43% | 31% | 42% | 49% | 47% | 41% | 41% | 47% | 49% |
| Closure of workplace | 36% | 53% | 51% | 39% | 8% | 36% | 35% | 37% | 33% | 37% | 34% | 59% | 40% | 32% | 35% | 34% | 41% | 38% | 35% | 33% | 40% | 46% |

**Figure 2** Bar chart showing how respondents from the following demographic groups were affected economically by COVID-19: at country level (TH, Thailand; MY, Malaysia; IT, Italy; SI, Slovenia), gender (M, male; F, female; O, other/prefer not to say); education level (P/S, primary or lower/secondary; T, tertiary); age (18–24 years old, 25–34 years old, 35–64 years old, 65+ years old); household size (1–4 people, ≥5 people); living with children under 18 years (Y, yes; N, no) and type of income (FBP, fixed/benefits/pension; CF, contract/freelance; O, other/no income).

the breakdown by country. All results in the following subsections are presented as weighted percentages.

## Views on economic impacts of COVID-19 and public health measures

In order to understand the economic impacts of COVID-19, respondents who had been working before the pandemic (paid or unpaid work) were asked whether COVID-19 had created any work-related inconvenience for them. Overall, 56% of respondents said that they experienced loss of earnings, 44% reduction of working hours, 36% closure of workplace and 14% job loss (figure 2, online supplemental table 2). A total of 75% reported that they continued to work during COVID-19. Of all respondents, 53% expressed financial concerns, and 32% worried about professional/career progression. Our results indicated that the most affected country was Thailand, with 85% of respondents reporting loss of earnings, 23% loss of job and 86% expressing financial concerns (online supplemental table 2). In contrast, fewer Slovenian respondents appeared to be affected economically, for example, 30% reported loss of earnings, 3% reported loss of job, and 27% had financial concerns.

To investigate the impact of public health measures on different social groups, we analysed responses based on gender, level of education, age group, household size, whether respondents lived with children under 18 years old and income type.

Overall, there were no significant differences between male, female and respondents who identified as 'other/ prefer not to say', and who had been working before COVID-19, in terms of loss of earnings, loss of job, reduction of working hours and closure of workplace (figure 2, online supplemental table 3). Overall, fewer women continued to work during COVID-19 (71% women vs 78% men; p=0.010). The trend was similar at country level, except for Malaysia (73% women vs 67% men; online supplemental table 3).

Overall, 65% of respondents with primary and secondary education who had been working before COVID-19 reported a loss of earnings, compared with 38% of respondents with tertiary education (p<0.001; figure 2, online supplemental table 4). More respondents with primary/secondary education lost their job (17% vs 8%; p<0.001), and had their working hours reduced (47% vs 37%; p<0.001). Fewer respondents with primary/secondary education continued to work (71%, vs 83%, p<0.001), and 59% reported financial concerns (vs 41%; p<0.001). This trend was mirrored at country level. Respondents with primary/secondary education were most affected in Thailand, where 90% reported loss of earnings, 24% reported loss of job, and 89% reported financial concerns (online supplemental table 4). Only 65% of respondents with primary/secondary education in Malaysia (vs 90% with tertiary education) and 59% in Italy (vs 79%) continued to work during COVID-19.

In order to assess whether age was a factor associated with economic impact, respondents were divided into four age groups in the analysis: 18–24-year olds, 25–34-year olds, 35–64-year olds and over 65-year olds (figure 2, online supplemental tables 5a–b). There were significant differences between age groups regarding loss of earnings (p=0.044): 67% of 65+ year olds reported loss of earnings, compared with 59% of 18–24-year olds, 47% of 25–34-year olds and 56% of 35–64-year olds. There were no significant differences overall regarding loss of job (p=0.053). However, the 18–24-year olds appeared to be most affected through reduction of working hours (p=0.016) and closure of workplace (p=<0.001). Only 54% of 18–24-year olds and 68% of 65+ year olds continued to work during COVID-19, compared with 78% of 25–34% and 78% of 35–64-year olds (p=0.<0.001). Analysing by country, the

18–24-year olds reported the higher job losses compared with the other groups in Thailand (32%), Malaysia (42%) and the UK (19%). Those over 65 years old were particularly affected in Italy, where 87% of 65+ year olds who had been working before COVID-19 reported loss of earnings, and 42% reported loss of job (n=12). In all countries, fewer 18–24-year olds continued to work during COVID-19, and in all countries except Thailand, fewer 65+ year olds continued to work during COVID-19.

Overall, more respondents living in larger households, and more respondents living with children under 18 in the household reported economic impacts (figure 2, online supplemental tables 6 and 7). Overall, 64% of respondents whose household included five people or more reported loss of earnings (compared with 53% of households with 1–4 people; p=0.003), and 20% reported loss of job (compared with 12%; p=0.005; online supplemental table 6). More respondents with children reported a loss of earnings compared with respondents without children (62% vs 53%; p=0.005), and higher job loss (18% vs 12%; p=0.008; online supplemental table 7). Analysing by country, respondents living with children appeared to be particularly affected in Thailand and Malaysia.

We also analysed responses according to three types of income: fixed income (eg, fixed salary, benefits or pension), flexible income (eg, contract, freelance) and other/no income (figure 2; online supplemental table 8). We did not ask for amount of income. Overall, respondents with fixed income were less affected economically than those with flexible or other/no income. Of the latter, only 38% reported loss of earnings, compared with 81% of respondents with flexible income and 69% of respondents with other/no income (p<0.001). Only 8% of people with fixed income had lost their job, compared with 22% with flexible income and 27% with other/ no income (p<0.001). At country level, the trends were similar (online supplemental table 8). Fewer people with flexible or other/no income continued to work in Malaysia (42% with flexible/25% with no/other income, compared with 83% with fixed income; p<0.001), UK (57%/62%, compared with 79%; p<0.001), Italy (51%/15%, compared with 81%; p<0.001) and Slovenia (57%/59%, compared with 84%; p<0.001).

## Views on social impacts of COVID-19 and public health measures

We asked respondents if they were concerned about the following areas of life if advised no physical contact/not allowed to go out/allowed to go out only for essential needs: caring responsibilities, physical health, recreational pursuits, sports, mental health and well-being, living arrangements, infrastructure (eg, access to transport, internet), social and religious and spiritual needs/aspects (online supplemental table 9). Overall, respondents expressed most concern about their social life (64%), their physical health (59%) and their mental health and well-being (58%). This trend was largely similar in individual countries, except for Thailand,

where caring responsibilities attracted the most concern (62%); Malaysia, where 58% were concerned about religion and spirituality and Slovenia, where 65% of people worried about recreational aspects. In general, there were no major differences between gender (online supplemental table 10), age groups (online supplemental table 11), education level (online supplemental table 12), household size (online supplemental table 13), living with children (online supplemental table 14) or income type (online supplemental table 15). However, two areas with the most significant differences between demographic groups were caring responsibilities and living arrangements. For example, 52% of women (compared with 42% of men and 46% of 'other/prefer not to say', p<0.001; online supplemental table 10), and 64% of those living with children under 18 (compared with 38% of those without children, p<0.001; online supplemental table 14) expressed concerns about caring responsibilities. Concerns about living arrangements were reported by 33% of those with primary/secondary education (compared with 26% with tertiary, p<0.001; online supplemental table 12), and 41% of those living in households with five or more people (compared with 28% in households with 1–4 people, p<0.001; online supplemental table 13). We asked respondents how many days they could cope with not going out except for essential needs/work, with answer options ranging from 1 to 59 days or more. In total, 44% of respondents said that they could cope for 29 days or longer (online supplemental table 16). However, coping time varied significantly between countries (p<0.001): in the UK, 60% of people felt they would be able to cope for 29 days or longer, whereas in Thailand, only 26% of respondents said that they could cope this long. Overall, gender and age did not appear to be associated with coping time (online supplemental tables 17 and 18). Factors that appeared to be associated with lower coping times were living in households with five or more people (p=0.023, online supplemental table 19), with children under 18 years (p=0.004, online supplemental table 20), having primary/secondary education (p<0.001, online supplemental table 21), and receiving flexible income (p<0.001; online supplemental table 22). Indicators varied at country level.

## Self-reported compliance and acceptance of public health measures

Next, we explored which factors were associated with compliance and agreement with public health measures. Of all respondents, 67% reported that they had changed their social behaviour *before* government restrictions were implemented (figure 3; online supplemental table 23). There were significant differences at country level (p<0.001): 93% of Thai respondents reported voluntary prerestriction behaviour change, followed by the UK (68%) and Malaysia (64%). Slovenian (47%) and Italian respondents (47%) reported the lowest levels of voluntary prerestriction behaviour change. Overall, 92% of respondents had used sanitizer products and alcohol, 82%

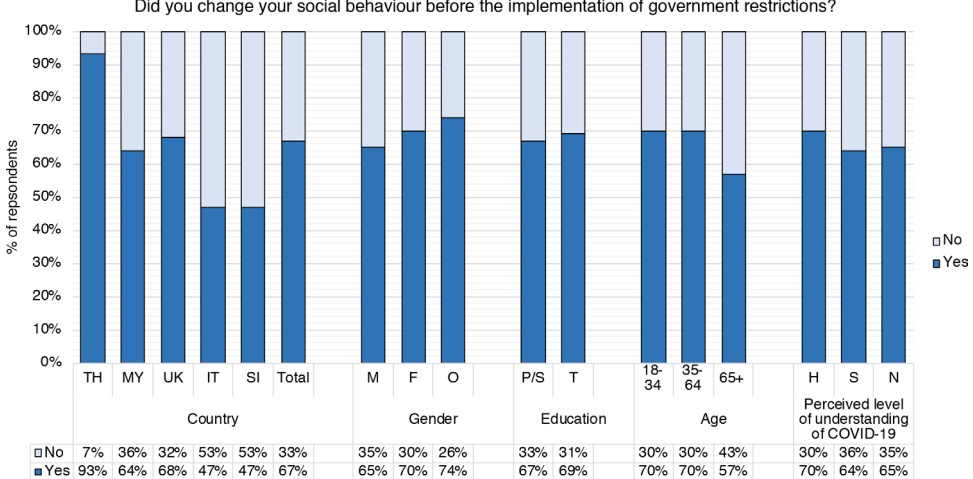

Did you change your social behaviour before the implementation of government restrictions?

| | TH | MY | UK | IT | SI | Total | | M | F | O | | P/S | T | | 18-34 | 35-64 | 65+ | | H | S | N |
|---|---|---|---|---|---|---|---|---|---|---|---|---|---|---|---|---|---|---|---|---|---|
| No | 7% | 36% | 32% | 53% | 53% | 33% | | 35% | 30% | 26% | | 33% | 31% | | 30% | 30% | 43% | | 30% | 36% | 35% |
| Yes | 93% | 64% | 68% | 47% | 47% | 67% | | 65% | 70% | 74% | | 67% | 69% | | 70% | 70% | 57% | | 70% | 64% | 65% |

**Figure 3** Breakdown of responses to the question 'Did you change your social behaviour before the implementation of government restrictions?' by country (TH, Thailand; MY, Malaysia; IT, Italy; SI, Slovenia) and demographic groups: gender (M, male; F, female; O, other/prefer not to say); education level (P/S, primary or lower/secondary; T, tertiary); age (18–34 years old, 35–64 years old, 65+ years old); self-reported/perceived level of understanding of COVID-19 (H, high/very high/expert level; S, some; N, a little/none at all).

avoided physical contact with anyone and 79% avoided physical contact with only vulnerable groups. In Thailand and Malaysia, 96% and 95% of respondents indicated that they had been using personal protective equipment (eg, face masks and gloves), compared with only 33% in UK, 55% in Italy and 67% in Slovenia (p<0.001). We also asked respondents how much they agreed with quarantine/isolation/social distancing measures and the statement that these are a necessary strategy to help control COVID-19 (online supplemental table 23). There was a significant difference between countries (p<0.001): agreement with public health measures was highest among respondents from Thailand (94%) and lowest among those from Slovenia (around 75%).

Overall, fewer male than female respondents changed their social behaviour before the government implemented official restrictions (65% and 70%, respectively, p=0.039; figure 3, online supplemental table 24). At country level, fewer men than women reported changing their social behaviour voluntarily, except in Thailand, where reported changes among men and women were similar (94%/92%, p=0.426). Overall, there were no significant differences between men and women when asked about how much they agreed with public health measures and the statement that these are a necessary strategy to help control COVID-19 (p=0.191; online supplemental table 24).

When it came to education level, there were no significant differences between respondents with primary/secondary and those with tertiary education regarding voluntary behaviour change before government-imposed restrictions (p=0.369), and agreement with public health measures and the statement that these are a necessary strategy to help control COVID-19 (p=0.304; figure 3, online supplemental table 25). Indicators varied at country level.

Overall, 70% of 18–34-year olds and 70% of 35–64-year olds indicated that they had voluntarily changed their behaviour before government restrictions, compared with only 57% of 65+year olds (p=0.004; figure 3, online supplemental table 26). This trend was similar at country level, except in Italy where 57% of 65+ year olds were most likely to change their behaviour, compared with 44% of 18–34% and 44% of 35–64-year olds. Overall, agreement with voluntary restrictions was similar across age groups (p=0.271; online supplemental table 26), but fewer 65+ year expressed agreement with restrictions that were government-enforced (p=0.003). Respondents over 65 years old in Slovenia reported the lowest agreement with the statement that quarantine/isolation/social distancing are a necessary strategy to help control COVID-19 (67%), compared with 96% in Thailand and 100% in Malaysia.

Finally, self-reported levels of understanding of COVID-19 did not significantly affect voluntary change of behaviour (p=0.091), or agreement with public health measures (p=0.688; online supplemental table 27).

## Self-perceived level of understanding of COVID-19

We asked respondents to indicate their perceived level of understanding of COVID-19. Overall, 59% of respondents indicated a 'high/very high' level of understanding, 36% reported 'some' understanding, and only 5% reported 'very little/none' (figure 4, online supplemental table 28). There were significant differences at country level (p<0.001): perceived levels of understanding were highest in Slovenia (66% reported 'high/very high', and 30% 'some' understanding) and Thailand (63% 'high/very high' and 33% 'some'), and lowest in Italy, with 47% reporting 'high/very high', and 50% reporting 'some' level of understanding.

To probe for factors associated with perceived level of understanding of COVID-19, we broke down responses

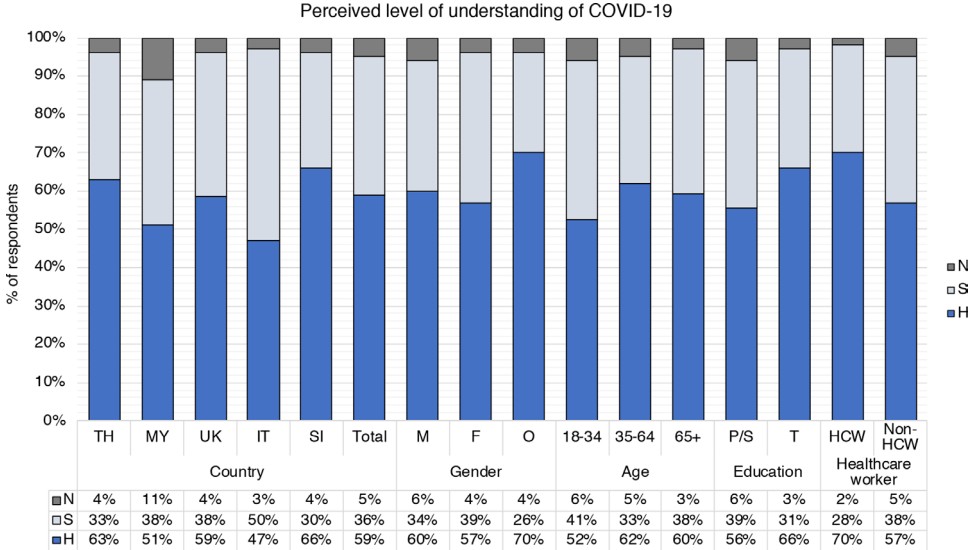

Perceived level of understanding of COVID-19

| | | | | | | | | | | | | | | |
|---|---|---|---|---|---|---|---|---|---|---|---|---|---|---|
| | **Country** | | | | | **Gender** | | | **Age** | | | **Education** | | **Healthcare worker** |
| | TH | MY | UK | IT | SI | Total | M | F | O | 18-34 | 35-64 | 65+ | P/S | T | HCW | Non-HCW |
| ■N | 4% | 11% | 4% | 3% | 4% | 5% | 6% | 4% | 4% | 6% | 5% | 3% | 6% | 3% | 2% | 5% |
| □S | 33% | 38% | 38% | 50% | 30% | 36% | 34% | 39% | 26% | 41% | 33% | 38% | 39% | 31% | 28% | 38% |
| ■H | 63% | 51% | 59% | 47% | 66% | 59% | 60% | 57% | 70% | 52% | 62% | 60% | 56% | 66% | 70% | 57% |

**Figure 4** Breakdown of responses to the question 'How would you rate your level understanding of the current quarantine/isolation/social distancing requirements for COVID-19?' Self-reported/perceived level of understanding of COVID-19 ((H, high/very high/expert level; S, some; N, a little/none at all) shown by country (TH, Thailand; MY, Malaysia; IT, Italy; SI, Slovenia) and demographic groups: gender (M,male; F, female; O, other/prefer not to say); age (18–34 years old, 35–64 years old, 65+ years old); education level (P/S, primary/secondary; T,tertiary); healthcare worker status (HCW, healthcare worker; Non-HCW, non-healthcare worker).

by gender, age, education and healthcare worker status (figure 4, online supplemental table 29). Overall, there was no significant difference between men, women and people who identified as other or preferred not to say (p=0.058; figure 4, online supplemental table 29). Age appeared to be a factor, as only 52% of 18–34-year old respondents self-reported 'high/very high' understanding compared with 62% of 35–64-year olds and 60% of 65+ year olds (p=0.033). Overall, fewer respondents with primary and secondary education self-reported 'high/very high' understanding (56% indicated 'high/very high' compared with 66% with tertiary education, p<0.001). Finally, healthcare worker status was associated with perceived higher understanding (p=0.001). This trend was similar at country level, except for Malaysia, where 49% of healthcare workers reported 'high/very high' understanding compared with 52% of non-healthcare workers (p=0.805) (online supplemental table 29).

Overall, higher levels of perceived understanding of COVID-19 were associated with higher levels of perceived understanding of public health measures (p<0.001; online supplemental table 30). For example, 88% of respondents who self-reported 'high/very high' understanding of COVID-19% and 50% who reported 'some' understanding felt that they had a 'high/very high' level of understanding of public health measures. In contrast, only 27% of respondents who reported 'very little/no' understanding of COVID-19 indicated a high understanding of public health measures.

## Information about COVID-19, unclear information and fake news

Throughout the study period, all five countries were running coordinated public information campaigns

(online supplemental figure 1).[32 33] When respondents were asked how they receive/received information about COVID-19 (online supplemental table 31), most reported traditional mass media (TV, radio, newspapers; 93%), followed by online methods (websites, email; 83%) and social media and messenger apps (79%). When asked about their preferences for receiving information, the top three responses were traditional mass media (78%), government or institution's website (77%) and online (76%). There were no significant differences based on gender (online supplemental table 32). Fewer respondents over 65 years said that they had used online channels or social media and messenger apps, and they expressed significantly lower preference for these channels too. For example, only 66% of over 65 year olds wanted to receive information online, compared with 78%/79% of the other age groups (p<0.001), and only 52% of over 65 year olds expressed preference for social media and messenger apps, compared with 64%/64% (p=0.005; online supplemental table 33). Overall, most respondents with primary/secondary education and those with tertiary education had received information through traditional mass media, and social media/messenger apps (online supplemental table 34). Fewer respondents with primary/secondary education had used online channels in the form of websites and emails (79% vs 92%, p<0.001), and more had received face-to-face information compared with those with tertiary education (43% vs 35%, p<0.001; online supplemental table 34). However, both education level groups indicated that their preferred methods of communication were mass media channels, online methods and government/institutions' websites.

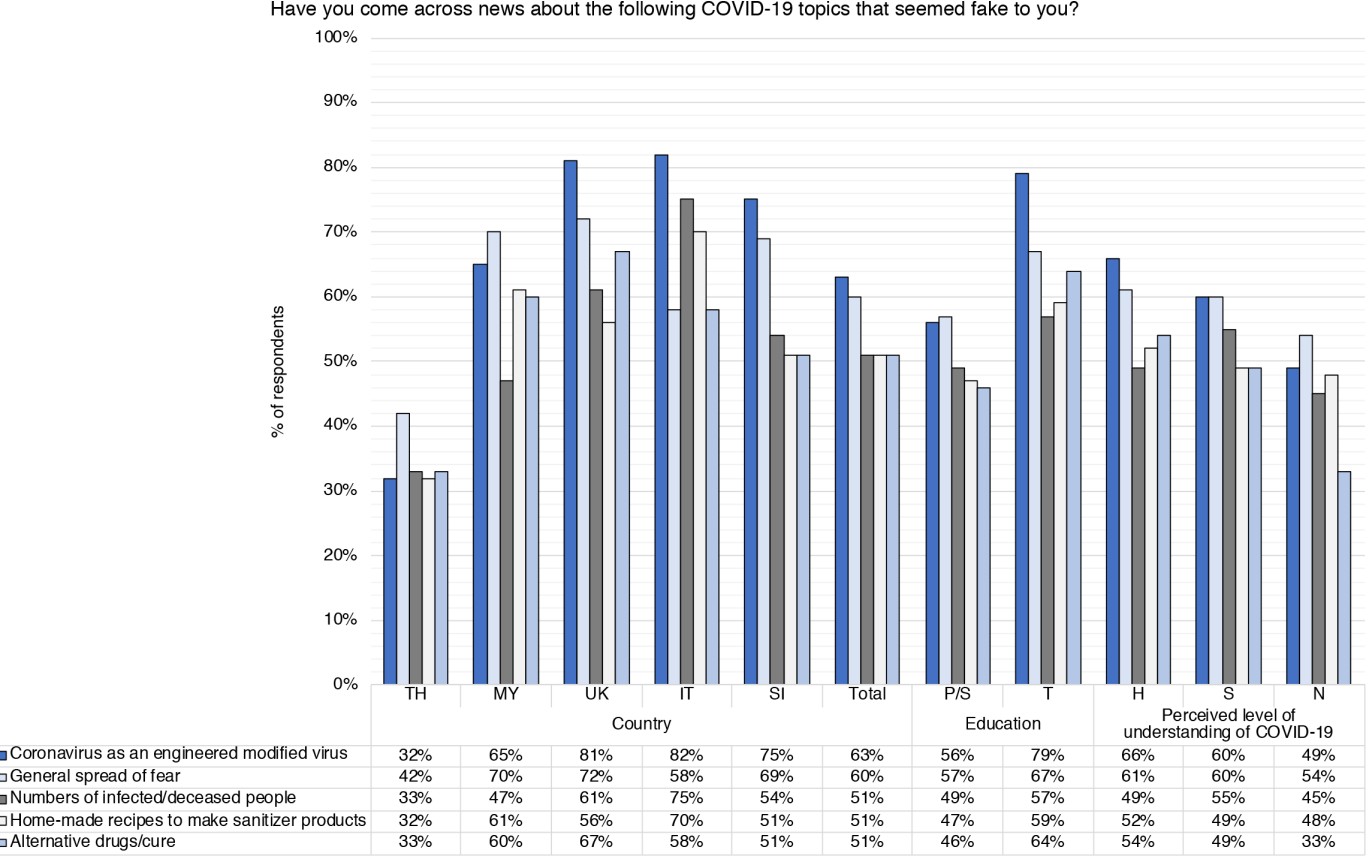

Have you come across news about the following COVID-19 topics that seemed fake to you?

| | TH | MY | UK | IT | SI | Total | P/S | T | H | S | N |
|---|---|---|---|---|---|---|---|---|---|---|---|
| ■ Coronavirus as an engineered modified virus | 32% | 65% | 81% | 82% | 75% | 63% | 56% | 79% | 66% | 60% | 49% |
| □ General spread of fear | 42% | 70% | 72% | 58% | 69% | 60% | 57% | 67% | 61% | 60% | 54% |
| ■ Numbers of infected/deceased people | 33% | 47% | 61% | 75% | 54% | 51% | 49% | 57% | 49% | 55% | 45% |
| □ Home-made recipes to make sanitizer products | 32% | 61% | 56% | 70% | 51% | 51% | 47% | 59% | 52% | 49% | 48% |
| ■ Alternative drugs/cure | 33% | 60% | 67% | 58% | 51% | 51% | 46% | 64% | 54% | 49% | 33% |

**Figure 5** Diagram showing how many survey respondents had come across five 'fake news' categories, in response to the question 'Have you come across news about the following COVID-19 topics that seemed fake to you?'. Breakdown by country (TH, Thailand; MY, Malaysia; IT, Italy; SI, Slovenia), gender (M, male; F, female; O, other/prefer not to say), age (18–34 years old, 35–64 years old, 65+ years old), education level (P/S, primary or lower/secondary; T, tertiary) and perceived level of understanding of COVID-19 (H, high/very high/expert level; S, some; N, a little/none at all).

We asked respondents if they had seen unclear or conflicting information about COVID-19 in nine categories relating to infection, symptoms and various public health measures. Overall, between 36% and 54% of respondents indicated that they had seen such information. Ways to avoid the infection (54%), government support schemes (52%) and testing (51%) were identified as the most unclear areas (online supplemental table 35). Thailand reported the lowest levels of seeing unclear or conflicting information in most categories (around 35%–40%), while respondents in the UK reported the highest levels in most categories (around 55%–70%). Overall, those with tertiary education reported significantly higher levels of seeing unclear information than those with primary/secondary education in almost all categories (p<0.001) except government support schemes (online supplemental table 36).

When asked 'Have you come across news about the following COVID-19 topics that seemed fake to you?', overall 63% of respondents had encountered news on 'Coronavirus as an engineered modified virus', 60% reported seeing 'general spread of fear' and 51% had come across seemingly fake news about 'numbers of infected/deceased people', 'home-made recipes to make sanitizer products' and 'alternative drugs/cure' (figure 5,

online supplemental table 35). Thailand reported the lowest percentages in all 'fake news' categories, with a range of 27%–42% (online supplemental table 35). Overall, respondents with tertiary education reported significantly higher levels of seeing 'fake news' in all categories compared with those with primary/secondary education (p<0.001; figure 5, online supplemental table 36). For example, only 56% of people with primary/secondary education reported coming across fake news about 'coronavirus as an engineered modified virus' versus 79% of those with tertiary education (p<0.001). There did not appear to be an association between self-reported levels of understanding of COVID-19 and seeing unclear/conflicting information or 'fake news' (online supplemental table 37).

## DISCUSSION

Our results indicate how public health measures that were in place between 1 May and 30 June 2020 affected a cohort of over 5000 respondents across five countries, and thus contribute new data and insights to these research areas.

## Groups most affected by COVID-19 public health measures

The following factors were associated with a negative economic impact: belonging to the age group 18–24 years or 65 and over, having lower education levels, living in larger households with five or more people, having children under 18 in the household and having flexible/no income. This suggests that COVID-19 public health measures can have greater negative impacts on already disadvantaged groups. Overall, it appeared that the 35–64-year old age group was less affected than other age groups. Possible explanations for this could be the types of sectors that younger and older people work in (eg, low paid or service industries),[34 35] or for older workers, shielding guidance issued by governments, lower levels of digital skills for remote working,[36] or discrimination in the form of ageism.[34 37] There were no significant differences between gender groups in our overall analysis. However, other studies have shown that COVID-19 has had a greater impact on women (eg, women are more likely to have temporary contracts[38 39] and disproportionally carry the burden of unpaid care).[40 41] A more detailed gender analysis to further break down our survey results is currently underway.

Our results showed that among the countries surveyed, respondents from Thailand reported the most adverse impacts. Thailand is a middle-income country with a large informal economy, and relies heavily on the tourism industry (15% GDP).[42] Thailand also had a high government SI during the period of the study (figure 1), which included closure of borders, businesses and nighttime curfews.[43] This meant that many informal street vendors and those working in the tourism industry (eg, tour operators) had no income or lost their jobs.

Overall, about two-thirds of respondents were most concerned about the effects of public health measures on their social life, their physical health and their mental health and well-being. These findings resonate with other studies showing the substantial negative impact of COVID-19 restrictions on mental health, well-being and social life.[44–46]

## Self-reported compliance and behavioural changes

A number of quantitative online surveys have examined experiences, knowledge, attitude and perceptions towards COVID-19 and public health measures, at country level,[38 47–56] and among different social groups.[57–60] Our findings show that self-reported compliance and behavioural change seemed to differ between countries. For example, respondents in Thailand indicated significantly higher levels of compliance, acceptance of public health measures and voluntary behavioural change compared with other countries. Although our survey was unable to implicate causality, it may contribute to better understanding of why Thailand has the lowest number of COVID-19 cases relative to its population among the countries who took part in the survey.[61] Some of our results with regard to gender and age were similar to trends reported in other studies. For example, results from a Hong Kong

study showed that female respondents, and those who reported higher levels of understanding of COVID-19, were more likely to adopt social distancing measures.[62] Similarly, a Chinese study found that men and those with a lower COVID-19 knowledge score were less likely to avoid crowded places or wear a mask outside.[51] Using survey data from 27 countries, Daoust[57] observed that compliance was not higher in older people even though they might be expected to comply more due to being a risk group. Similarly, our data showed that overall and in Malaysia, UK and Slovenia, far fewer respondents over 65 years reported changing their behaviour voluntarily before official restrictions came into place. However, overall, over 80% of respondents in all three age groups expressed agreement when asked if they would comply voluntarily or with government-mandated restrictions (online supplemental table 26).

## Improving COVID-19 communication

Our findings indicated that younger age and lower education levels appeared to be associated with lower self-perceived/subjective levels of understanding of COVID-19. Also, higher self-reported levels of understanding of COVID-19 seemed to be associated with higher self-perceived levels of understanding of public health measures. A recent modelling study suggests that self-imposed public health measures combined with fast spreading of disease awareness in the population can help reduce transmission of the virus.[11] Our findings suggest that specific groups of people, such as those with primary/secondary education levels and those 18–34-year old, may benefit most from targeted COVID-19 communication initiatives.

In terms of channels of communications, the three most popular channels across countries were traditional mass media, government or institutional websites and online media. Similar results emerged from a recent survey carried out in the Netherlands, Germany and Italy.[54] However, respondents in Thailand reported that they preferred to receive information face-to-face, especially those with primary/secondary education. This suggests that in order for communication strategies to be effective, they need to be sensitive to population preferences and tailored to local contextual factors (eg, levels of connectivity, literacy.[63]

Our survey showed that a significant proportion of the population received conflicting information and news that seemed fake to them, in particular about coronavirus being an engineered modified virus. These findings confirm other reports that misinformation and what has been termed the COVID-19 'infodemic' is widespread.[58 64 65] More efforts should be made to curb misinformation and disinformation, taking into account the needs of different groups.[46]

## Strengths and limitations

Our online survey enabled us to capture people's experiences and concerns in multiple domains, in five countries,

all of which had restrictions in place, during the relatively early stage of the COVID-19 pandemic. To the best of our knowledge, the SEBCOV study was one of the largest international mixed-methods studies conducted on the impact of COVID-19. To maximise the number of respondents and the likelihood of getting honest answers, the survey was completely anonymous. Due to the relatively large sample of respondents in each country, we were able to compare population segments (eg, men vs women or younger vs older people) in our overall cohort and at country level. We did not aim to obtain nationally representative samples and acknowledge that although we used weighting strategies in our analysis, our results may not be fully representative of the populations in the respective countries. Similarly, there might be differences in the frequency of demographic groups (eg, 18–24 years old who had been working before COVID-19) between the different countries, which might affect the interpretation of our data at country level. Overall, there was a high proportion of respondents who were healthcare workers (19%), and some variation in this proportion between countries. This may have influenced the country-level analysis, in particular in the areas of perceived understanding, compliance/agreement and communication preferences.

Because the survey was online, only people who were literate, had internet access and had access to computers or smartphones could take part. Due to COVID-19 related restrictions, it was not possible to conduct face-to-face data collection to reach groups who were illiterate in the language of the survey, or who did not have access to online technology. This is likely to have biased our data towards more educated and economically advantaged populations. Our study was also subject to response bias and other biases arising from self-reporting and recall. Our study was designed to capture the views and perceptions of respondents on how COVID-19 impacted them socially and economically rather than standard social and economic impact indicator, which are captured by other studies. Similarly, our survey captured perceived level of understanding of COVID-19 and public health measures rather than actual level of understanding. We were able to identify communication needs and preferences of our respondents, which can be used as guidance for organisations running public health communication initiatives. As the media landscapes vary among countries, other factors like freedom of press or the proportion of digital media consumption are likely to influence people's responses. Finally, as a cross-sectional survey, our data only sheds light on the prevalence of certain phenomena and opinions of respondents but does not imply causality.

The results of the survey reported here form part of a mixed-methods study, which also includes an in-depth qualitative study, the findings of which are currently being analysed and will be published separately. Combined, our results may help explain some of the trends reported in this survey, as well as the differences between countries and social groups. We have also conducted a preliminary analysis of unweighted Thai survey responses during May 2020, which includes more detailed breakdowns by regions within Thailand.[66]

## CONCLUSION

Our data confirmed that COVID-19 and public health measures have unequal effects on different countries and different social groups within countries. As such, this study helps to expose some of the social and economic inequalities resulting from COVID-19 and public health measures, and contributes to an important body of research showing that NPIs have a greater impact on those who are socioeconomically disadvantaged. Our findings provide an indication of the social groups who may be most in need of support during pandemics, so that existing social inequalities are not perpetuated and worsened. Finally, our data can help to inform future strategies for effective communication in order to mitigate the impacts of COVID-19.

**Author affiliations**
[1]Mahidol Oxford Tropical Medicine Research Unit, Faculty of Tropical Medicine, Mahidol University, Bangkok, Thailand
[2]Centre for Tropical Medicine and Global Health, Nuffield Department of Medicine, University of Oxford, Oxford, UK
[3]Paediatric Ethics Committee; Research Ethics Committee, University Hospital of Padua, Padua, Italy
[4]Department of Tropical Hygiene, Faculty of Tropical Medicine, Mahidol University, Bangkok, Thailand
[5]Faculty of Arts and Social Science, Universiti Tunku Abdul Rahman, Kampar, Malaysia
[6]Emergency and Trauma Department, Sabah Women and Children's Hospital, Ministry of Health, Kota Kinabalu, Malaysia
[7]Emergency Department, Loh Guan Lye Specialists Centre, Georgetown, Malaysia
[8]Department of Statistical Sciences, University of Padua, Padua, Italy
[9]Faculty of Medicine, University of Ljubljana, Ljubljana, Slovenia
[10]Department of Radiation Oncology, Institute of Oncology, Ljubljana, Slovenia
[11]Institute for Social Studies, Science and Research Centre Koper, Koper, Slovenia
[12]Department of Endocrinology, Diabetes and Metabolic Diseases, University Children's Hospital, University Medical Center Ljubljana, Ljubljana, Slovenia
[13]The Ethox Centre, Nuffield Department of Population Health, University of Oxford, Oxford, UK
[14]Institute for Philosophical Studies, Science and Research Centre Koper, Koper, Slovenia
[15]Department of Global Health and Development, London School of Hygiene and Tropical Medicine, London, UK
[16]Emergency and Trauma Department, Queen Elizabeth Hospital, Ministry of Health, Kota Kinabalu, Malaysia

**Acknowledgements** We would like to thank all participants who took part in our study. We also thank the people who helped to test and promote the survey, and the members of the Bangkok Health Research and Ethics Group for their input into the study. We would like to thank the SoNAR-Global Network for the social science expertise in the development of this project and the ODK core team for their input in design of the quantitative survey. Finally, we would like to thank Prof Nicholas P J Day, Prof Richard Maude and Dr Tamara Giles-Vernick for critical reading of the manuscript.

**Contributors** AO and PYC oversaw the whole project and wrote the initial draft of the manuscript. AO, GC, WP-N, PKC, P-KC, MS, MLS, TP, NW, S-aA, BN, SR, NK, DO, RC and PYC developed the survey and translations. AO, GC, WP-N, P-KC and LS led the project in the UK, Italy, Thailand, Malaysia and Slovenia, respectively. MM and PP conducted the statistical analysis, figures and tables, with critical input from MS, AO and PYC. MLS critically reviewed the manuscript, figures and tables. AO,

GC, WP-N, PKC, P-KC, MLS, MO, KP, UG, MLS, TP, S-aA, BN, SR, LS, NK, CRSM-Y, DO, RC and PYC implemented the survey in their respective countries. All authors contributed to the draft paper, and approved the final version of the paper. PYC conceived the project and is the guarantor of the paper.

**Funding** This research was funded by a Wellcome Trust Institutional Translational Partnership Award (iTPA), Thailand (210559), and a Wellcome Trust Strategic Award (096527). The Mahidol Oxford Tropical Medicine Research Unit is funded by the Wellcome Trust (106698 and 220221). This study was also supported by the Sonar-Global project which has received funding from the European Union Horizon 2020 Research and Innovation Program (825671). The research programme in Slovenia is funded by the Slovenian Research Agency (Javna agencija za raziskovalno dejavnost RS) (P6-0279).

**Competing interests** None declared.

**Patient consent for publication** Obtained.

**Ethics approval** Ethics approval was granted by Oxford Tropical Research Ethics Committee (OxTREC, reference no.520-20), covering all countries; the Faculty of Tropical Medicine Ethics Committee, Thailand (FTMEC, ref: MUTM 2020-031-01); the Medical Research and Ethics Committee (MREC), Ministry of Health Malaysia (MOH), Malaysia, ref: NMRR-20-595-54437 (IIR), the Universiti Tunku Abdul Rahman (Utar) Scientific and Ethical Review Committee (SERC, ref: (U/SERC/63/2020), Malaysia and the National Medical Ethics Committee of the Republic of Slovenia (0120-237/2020/7). Additional ethics committee approval from Italy was not required for the study to be conducted there.

**Provenance and peer review** Not commissioned; externally peer reviewed.

**Data availability statement** Data are available upon reasonable request. All authors recognise the value of sharing individual level data. We aim to ensure that data generated from all our research are collected, curated, managed and shared in a way that maximises their benefit. Data underlying this publication are available upon request to the Mahidol Oxford Tropical Medicine Research Uni Data Access Committee at https://www.tropmedres.ac/units/moru-bangkok/bioethics-engagement/data-sharing.

**ORCID iDs**
Anne Osterrieder http://orcid.org/0000-0003-3378-4211
Giulia Cuman http://orcid.org/0000-0001-6243-8487
Wirichada Pan-Ngum http://orcid.org/0000-0002-9839-5359
Ksenija Perkovic http://orcid.org/0000-0001-9778-8436
Urh Groselj http://orcid.org/0000-0002-5246-9869
Lenart Skof http://orcid.org/0000-0003-2199-3550
Constance R S Mackworth-Young http://orcid.org/0000-0002-9725-7931
Phaik Yeong Cheah http://orcid.org/0000-0001-6327-3266

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
