## [Reviewer comments · BMJ Open]

ARTICLE DETAILS

TITLE (PROVISIONAL)	Economic and social impacts of COVID-19 and public health measures: results from an anonymous online survey in Thailand, Malaysia, the United Kingdom, Italy and Slovenia
AUTHORS	Osterrieder, Anne; Cuman, Giulia; Pan-Ngum, Wirichada; Cheah, Phaik Kin; Cheah, Phee-Kheng; Peerawaranun, Pimnara; Silan, Margherita; Orazem, Miha; Perkovic, Ksenija; Groselj, Urh; Schneiders, Mira Leonie; Poomchaichote, Tassawan; Waitkira, Naomi; Asarath, Supa-at; Naemiratch, Bhensri; Ruangajorn, Supanat; Skof, Lenart; Kulpijit, Natinee; Mackworth-Young, Constance; Ongkili, Darlene; Chanviriyavuth, Rita; Mukaka, Mavuto; Cheah, Phaikyeong

VERSION 1 – REVIEW

REVIEWER	Valente de Almeida, Sara Universidade Nova de Lisboa Nova School of Business and Economics
REVIEW RETURNED	16-Jan-2021

GENERAL COMMENTS	Dear Authors, Thank you for your submission. I find the paper to be well written, organised and easy to read. In addition, although mainly descriptive and with a restrictive sample, it contains relevant information for the general public and policy implications. I do have some minor comments/changes: - Either use % or "percent" when describing percentages through the paper;- Concepts as "least severe economic impacts" are not clear. What is considered severe? This concept may change according to the country.- 18-34 age group is a bit large to study labour market consequences is a bit wide, depending on the country. I would imagine that the percentage of young adults (with 18) working is much higher in Thailand than in the UK. This takes me to another point with is, people from the same age group in such different countries will probably have different percentages (maybe add to the limitations).- Using the benchmark "households with 6 or more members", as shown in your tables, gives much more information on Thailand
---

	than on Italy. Even though you use weights, I would recommend to lower the threshold and see if there is more balance between countries for this variable. - When you use the variable "Level of understanding" based on self-reported measures this is extremely biased by culture, age, context, income level, education (as you know). I believe in your limitations you should talk about this variable in particular since it has a lot of focus in your analysis. For the future, I would suggest to complement this question with some objective measures of understanding (maybe ask about how the virus spreads with multiple choice). Otherwise, this variable maybe an indicator of self confidence rather than understanding of the virus. (maybe add some references arguing for the accuracy of this question? if it exists) - Finally, I would welcome more context on the information channels of each country. This said, in section "Information about COVID-19, unclear information and fake news" I would appreciate some more information about how each country communicates, how much information does the population really has, is the media controlled by the government in some countries, how frequent? does every country has daily reports? This is important to interpret the results, including levels of compliance and confidence in the government. Thank you, Sara Valente de Almeida
--	---

REVIEWER	Saghafipour, Abedin Qom Univ Med Sci
REVIEW RETURNED	27-Jan-2021

GENERAL COMMENTS	The reviewer also provided a marked copy with additional comments. Please contact the publisher for full details.
---

VERSION 1 – AUTHOR RESPONSE

Response to reviewers

We would like to thank the reviewers for taking the time to read our manuscript, and for their constructive comments, which have helped to improve the manuscript. Please find our responses below.

Responses to Reviewer 1

- 1) Either use % or "percent" when describing percentages through the paper;

We have revised the text and now use % consistently throughout the text.

- 2) Concepts as "least severe economic impacts" are not clear. What is considered severe? This concept may change according to the country.

We agree with this point. We used these phrases e.g. to express that fewer Slovenian respondents reported loss of job/loss of earnings compared to the overall cohort and the other countries. We have

changed the manuscript to rephrase statements like “least severe impacts” or “worse economic impacts” (please see tracked changes).

- 3) 18-34 age group is a bit large to study labour market consequences is a bit wide, depending on the country. I would imagine that the percentage of young adults (with 18) working is much higher in Thailand than in the UK. This takes me to another point with is, people from the same age group in such different countries will probably have different percentages (maybe add to the limitations).

This is a fair point. We have re-analysed our data, breaking down the age group 18-34 years into 18-24 (final school years/University/junior professionals) and 25-34 (early career professionals). In our survey, we collected data on age in year groups (18-24, 25-34 etc), instead of individual ages, and therefore unfortunately we do not have data for 18 year olds only. We have updated Suppl. Table 1. and split up Suppl. Table 5 into 5a and 5b.

- 4) Using the benchmark "households with 6 or more members", as shown in your tables, gives much more information on Thailand than on Italy. Even though you use weights, I would recommend to lower the threshold and see if there is more balance between countries for this variable.

We have now reanalysed the data using “households with 1-4 members” and “households with 5 or more members”. We chose 5 as the new cut-off for “large” households to reflect multigenerational households commonly found in Thailand and Malaysia. As result of the re-analysis, we also revised the paragraph on page 10 to improve clarity and presentation of our findings (e.g. ‘living arrangements’ was one area that larger households were significantly more concerned about).

- 5) When you use the variable "Level of understanding" based on self-reported measures this is extremely biased by culture, age, context, income level, education (as you know). I believe in your limitations you should talk about this variable in particular since it has a lot of focus in your analysis. For the future, I would suggest to complement this question with some objective measures of understanding (maybe ask about how the virus spreads with multiple choice). Otherwise, this variable maybe an indicator of self confidence rather than understanding of the virus. (maybe add some references arguing for the accuracy of this question? if it exists)

Thank you for this comment. Our intention was to determine peoples’ own perception, as we think this will affect their behaviour. We did not include knowledge-based questions because it may deter some people from answering the survey. Also, at the time of the survey, there were many unknowns in COVID-19, and scientists were still actively determining its origins, how it spreads etc. It would have been challenging to design quiz-like questions then. Nevertheless, we have added this point in the limitation section. We have also added “perceived” in the relevant subtitle. It now reads “Self-perceived level of understanding of COVID-19”.

- 6) Finally, I would welcome more context on the information channels of each country. This said, in section "Information about COVID-19, unclear information and fake news" I would appreciate some more information about how each country communicates, how much information does the population really has, is the media controlled by the government in some countries, how frequent? does every country has daily reports? This is important to interpret the results, including levels of compliance and confidence in the government.

This is a valid point. We did some research on government communications during COVID-19. We found a few studies, which looked at media consumption of people during COVID-19 in more detail – usually either at individual country level, or with a few selected countries [1, 2]. To our knowledge, the Coronavirus Government Response Tracker, visualised via ‘Our World in Data’ [3], is the only aggregator that collected data on government/public health information campaigns from all countries throughout the course of the whole pandemic. We have included information on the coordination level of public health campaigns in form of a supplementary Figure 1. It shows that all five countries had coordinated public health campaigns from 1st March throughout the time period of our study.

We agree with the reviewer’s comments about the potential connection between countries’ media landscapes and people’s compliance and trust. However, we feel that this is out of scope for our study due to the complexity of the analysis needed, and because we are public engagement/science communication experts, not media/journalism experts. For example, when preparing our response to the reviewer’s comments, we identified the levels of press freedom for each country, as indicated by the World Press Freedom Index (see Table 1 below). Malaysia and Thailand have a lower rank and a higher score compared to the European countries. Both countries have also managed the pandemic better than the European countries. Correlating compliance and confidence in government with the media landscape in each country is complex, and is being addressed in other studies [e.g. 1, 2].

Our main aim in this study was to capture people’s communication needs and preferences, to provide guidance for organisations running public health communication campaigns, and to help improve communication efforts.

To acknowledge the reviewer’s point, we have included the following sentences in the ‘Strengths and limitations’ section: “We were able to identify communication needs and preferences of our respondents, which can be used as guidance for organisations running public health communication initiatives. As the media landscapes vary among countries, other factors like freedom of press or the proportion of digital media consumption are likely to influence people’s responses.”

Table 1: Levels of freedom available to journalists in the five study countries, ranked according to the 2020 World Press Freedom Index [4].

Country	Rank (out of 180)	Score (out of 100, with 0 being the best possible and 100 the worst)
Thailand	144	44.94
Malaysia	101	33.12
United Kingdom	35	22.93
Italy	41	23.69
Slovenia	32	22.64

References:

[1] Sjölander-Lindqvist, A., Larsson, S., Fava, N., Gillberg, N., Marciandò, C., & Cinque, S. (2020). Communicating about COVID-19 in four European countries: similarities and differences in national discourses in Germany, Italy, Spain, and Sweden. *Frontiers in Communication* 5(97). doi:10.3389/fcomm.2020.593325

[2] Tench, R. "Communication channels, trust and messaging about COVID-19: experiences of populations in three key European countries, UK, Italy and Spain" [Available from: <https://www.leedsbeckett.ac.uk/blogs/lbu-together/2020/06/communication-channels-trust-and-messaging-about-covid19/>]

[3] COVID-19: Public Information Campaigns. Available at: <https://ourworldindata.org/covid-public-information-campaigns>

[4] Reporters Without Borders (2021). World Press Freedom Index. [Available from: <https://rsf.org/en/ranking>]

Reviewer 2

1) Text suggestion: "economic and social..."

The study was registered under the title "SEBCOV – Social, ethical and behavioural aspects of COVID-19" (e.g. see Pan-Ngum W., Poomchaichote T., Cuman G., et al. Social, ethical and behavioural aspects of COVID-19 [version 2; peer review: 2 approved]. Wellcome Open Res 2020;5(90) doi: <https://doi.org/10.12688/wellcomeopenres.15813.2>; <https://covid19crc.org/covid-19-studies/sebcov/>). Therefore, to avoid confusion, we would prefer to not change the name in this instance.

2) "Statistical analysis?"

We changed the title of this abstract section to "Study design and statistical analysis" and added more information about the statistical analysis.

3) "() for all reports is necessary"

We assumed that these comments indicated where percentages should be included in the abstract. We struggled to address this comment, as we do not have single aggregated percentage values for economic or social impacts. For example, 'greater negative impacts' summarises the differences in loss of earnings, loss of job, and whether people continued to work during COVID-19. The word count does not allow us to go into this much detail, and therefore we attempted to summarise the trends observed in our data. We appreciate that the text should be consistent and removed the following sentence part: "...with 60% and 26% respectively able to cope with 29 days or longer." We would welcome guidance by the Editor on this matter, and whether we should remove percentages all together, or include them for selected results (e.g. social impacts, days of coping).

4) "full name of COVID-19 is proposed."

We included the full name, 'coronavirus disease 2019 (COVID-19) in the abstract and introduction.

5) (WHO)

Included in the text.

6) "ethical????"

See response to point 1 above.

7) “at first study area “

We have now included the study area.

8) “population of these countries”

We have included the population of these countries in the section ‘Study area’, in Methods

9) “How did you calculate the sample size?”

The following text was added to the section “Sample size”:

The following sample size formula was used $n = \frac{Z_{1-\alpha/2}^2 P(1-P)}{d^2}$ where P is the anticipated prevalence, d is the margin of error, $Z_{1-\alpha/2}$ is the standard normal value corresponding to the upper tail probability of $\alpha/2$, $\alpha=0.05$ (for a 95% confidence interval), n is the sample size.

10) (...%) for all results is necessary.

We have included percentages in the main text and in Supplementary Table 1.

11) Economic impacts of COVID-19 and public health measures should be calculate in before and after study.

Our study was designed to capture the views and perceptions of respondents on how COVID-19 impacted them socially and economically rather than actual social and economic impacts which was captured by other means such as by government agencies. We have included this in the limitations section. We have also edited the subtitles in the Results section to reflect this, i.e. “Views on economic impacts of COVID-19 and public health measures” and “Views on social impacts of COVID-19 and public health measures”. Lastly, we added this limitation to the article summary at the start.

12) Social impacts of COVID-19 and public health measures should be calculate in before and after study.

As above.

13) “we do not have question sentence in the discussion.”

We have revised it to read “Groups most affected by COVID-19 public health measures”.

14) “you should revised the conclusion. Conclusions should be written based on the knowledge of the authors themselves and you have no right to refer to this section.”

15) “Conclusions should be written based on the results of the present study.”

The Conclusions section has been revised.

Other changes to the text

- Changed sentence in the abstract and introduction (2nd paragraph) to: “In the absence of widely available vaccines and pharmaceutical treatments...” (as vaccines are now being rolled out).
- Added clarification in the introduction (p4): “These data can be used to supplement quantitative/statistical data on economic and social impacts to provide a better understanding of the effects of COVID-19 and its related public health measures.”
- Summary section: we deleted one of the bullet points to keep the total number to 5.

In addition to the above, we have made minor changes throughout the text to improve its clarity.

As a result of responding to the reviewers’ comments, our word count is now 5757.

VERSION 2 – REVIEW

REVIEWER	Valente de Almeida, Sara Universidade Nova de Lisboa Nova School of Business and Economics
REVIEW RETURNED	29-Mar-2021

GENERAL COMMENTS	The authors addressed my previous comments very thoroughly. I find the paper quite interesting and with relevant and updated information.
---